# The Role of Tumor Necrosis Factor Alpha (TNF-α) in Autoimmune Disease and Current TNF-α Inhibitors in Therapeutics

**DOI:** 10.3390/ijms22052719

**Published:** 2021-03-08

**Authors:** Dan-in Jang, A-Hyeon Lee, Hye-Yoon Shin, Hyo-Ryeong Song, Jong-Hwi Park, Tae-Bong Kang, Sang-Ryong Lee, Seung-Hoon Yang

**Affiliations:** 1Department of Medical Biotechnology, Collage of Life Science and Biotechnology, Dongguk University, Seoul 04620, Korea; sera1586@naver.com (D.-i.J.); skandkgus@dgu.ac.kr (A.-H.L.); rtrt45689@naver.com (H.-R.S.); whdgnl7576@gmail.com (J.-H.P.); 2School of Life Science, Handong Global University, Pohang, Gyeongbuk 37554, Korea; 21900409@handong.edu; 3Department of Pharmacy, College of Pharmacy, Yonsei University, Seoul 03722, Korea; 4Department of Biotechnology, College of Biomedical and Health Science, Konkuk University, Chungju 27478, Korea; kangtbko@kku.ac.kr; 5Department of Biological Environmental Science, Collage of Life Science and Biotechnology, Dongguk University, Seoul 04620, Korea

**Keywords:** TNF-α, autoimmune diseases, rheumatoid arthritis, inflammatory bowel disease, psoriatic arthritis, TNF-α inhibitors

## Abstract

Tumor necrosis factor alpha (TNF-α) was initially recognized as a factor that causes the necrosis of tumors, but it has been recently identified to have additional important functions as a pathological component of autoimmune diseases. TNF-α binds to two different receptors, which initiate signal transduction pathways. These pathways lead to various cellular responses, including cell survival, differentiation, and proliferation. However, the inappropriate or excessive activation of TNF-α signaling is associated with chronic inflammation and can eventually lead to the development of pathological complications such as autoimmune diseases. Understanding of the TNF-α signaling mechanism has been expanded and applied for the treatment of immune diseases, which has resulted in the development of effective therapeutic tools, including TNF-α inhibitors. Currently, clinically approved TNF-α inhibitors have shown noticeable potency in a variety of autoimmune diseases, and novel TNF-α signaling inhibitors are being clinically evaluated. In this review, we briefly introduce the impact of TNF-α signaling on autoimmune diseases and its inhibitors, which are used as therapeutic agents against autoimmune diseases.

## 1. Introduction

Tumor necrosis factor alpha (TNF-α) is a cytokine that has pleiotropic effects on various cell types. It has been identified as a major regulator of inflammatory responses and is known to be involved in the pathogenesis of some inflammatory and autoimmune diseases [1]. Structurally, TNF-α is a homotrimer protein consisting of 157 amino acids, mainly generated by activated macrophages, T-lymphocytes, and natural killer cells [2]. It is functionally known to trigger a series of various inflammatory molecules, including other cytokines and chemokines. TNF-α exists in a soluble and transmembrane form. The transmembrane TNF-α (tmTNF-α) is the initially synthesized precursor form and is required to be processed by TNF-α-converting enzyme (TACE), a membrane-bound disintegrin metalloproteinase, to be released as the soluble TNF-α (sTNF-α) [3]. The processed sTNF-α then facilitates various biological activities through type 1 receptors (TNFR1, also known as TNFRSF1A, CD120a, and p55) and type 2 receptors (TNFR2, also known as TNFRSF1B, CD120b, and p75) [4,5,6,7]. tmTNF-α also acts on both TNFR1 and TNFR2, but its biological activities are expected to be mediated mainly through TNFR2 [8]. TNFR1 is expressed by all human tissues and is the key signaling receptor for TNF-α. TNFR2 is generally expressed in immune cells and facilitates limited biological responses [9]. In general, TNF-α binds to its receptors, mainly TNFR1 and TNFR2, and then transmits molecular signals for biological functions such as inflammation and cell death. TNFR1 is activated by both sTNF-α and tmTNF-α, and it processes a death domain (DD) that interacts with the TNFR1-associated death domain (TRADD) adaptor protein [10].

The activation of TNFR1 can trigger the formation of different signaling complexes, referred to as complexes I, IIa, IIb, and IIc, which result in distinct cellular responses [11,12]. During the assembly of complex I, the activated TNFR1 binds to TRADD, followed by the association and interaction of various components, including receptor-interacting serine/threonine-protein kinase 1 (RIPK1), TNFR-associated factor 2 or 5 (TRAF2/5), cellular inhibitor of apoptosis protein 1 or 2 (cIAP1/2), and the linear ubiquitin chain assembly complex (LUBAC) [13]. This signaling pathway results in the activation of nuclear factor κB (NF-κB) and mitogen-activated protein kinases (MAPKs) [14,15]. The functional outcome of complex I signaling is known to be the induction of inflammation, tissue, cell survival and proliferation, and the immune defense against pathogens [12,16]. Unlike complex I, which is assembled at the plasma membrane, complexes IIa, IIb, and IIc are assembled in the cytoplasm [17]. Complex IIa consists of TRADD, RIPK1, TRAF2, cIAP1/2, pro-Caspase-8, and Fas-associated protein with death domain (FADD) [18,19]. Complex IIb has the same composition as complex IIa with the addition of RIPK3 [18]. The formation of complexes IIa and IIb (also known as apoptosome) activates caspase-8 and results in apoptosis. Complex IIc (known as necrosome) is known to be formed by RIPK1 and RIPK3, which bind to form the complex when they remain uncleaved. Complex IIc activates the mixed lineage kinase domain-like protein (MLKL) via RIPK3-mediated phosphorylation, which induces necroptosis and inflammation (Figure 1) [13,20]. 

TNFR2 is suggested to be fully activated by tmTNF-α, primarily in the context of cell-to-cell interactions. Unlike TNFR1, TNFR2 lacks a death domain and thus is unable to induce programmed cell death directly (Figure 2) [8]. During assembly, TNFR2 recruits TRAF2 along with TRAF1, cIAP1, and cIAP2, and this complex allows the downstream activation of NF-κB, MAPKs, and protein kinase B (AKT) [17]. Functionally, TNFR2 is mainly associated with homeostatic bioactivities, including tissue regeneration, cell proliferation, and cell survival [21]. However, this pathway is also known to trigger inflammatory responses and host defense against pathogens. Overall, TNFR1 is essential to induce cytotoxic and proinflammatory TNF-α responses, while TNFR2 may largely mediate cell activation, migration, or proliferation (Figure 1).

Physiologically, TNF-α is a crucial component for a normal immune response. TNF-α can activate the immune system to regulate; however, the inappropriate or excessive production of TNF-α can be harmful and may lead to disease. Rheumatoid arthritis (RA) [22], inflammatory bowel disease (IBD) [23,24,25], psoriatic arthritis (PsA), psoriasis (PS) [26], and noninfectious uveitis (NIU) are induced by the abnormal secretion of TNF-α; thus, TNF-α can be classified as a key factor in the pathological development.

Due to the involvement of TNF-α in the pathogenesis of autoimmune diseases, TNF-α inhibitors have been successfully developed and applied in the clinical treatment of autoimmune diseases such as Crohn’s disease (CD) and RA [27,28]. Therapeutic drugs act as antagonists by blocking the interaction of TNF-α with TNFR1/2 or, in some cases, as agonists by stimulating reverse signaling, causing the apoptosis of TNF-α producing immune cells [29,30,31]. Several TNF-α inhibitors have been approved for clinical use: etanercept, infliximab, adalimumab, golimumab, and certolizumab. The impact of TNF-α signaling on each type of autoimmune disease will be introduced in this review, as well as the evaluation of current TNF-α inhibitors utilized as therapeutic drugs against autoimmune diseases.

## 2. TNF-α Signaling in Autoimmune Diseases

### 2.1. Rheumatoid Arthritis

Rheumatoid arthritis (RA) is a chronic autoimmune disorder that primarily affects the joints and typically results in redness, swelling, and arthralgia. In serious cases, it limits the range of motion [32]. RA affects nearly 1% of the population, and it is characterized by the inflammation of synovial tissue, leading to progressive damage, including the erosion of adjacent cartilage and bone, which can lead to chronic disability [33]. TNF-α is considered the major inflammatory cytokine involved in the pathogenesis of RA and is found in high frequencies in patients with the disease. Inflammation is associated with the accumulation of inflammatory cells, predominantly type 1 helper T cells (Th1) and macrophages but also B cells, plasma cells, and dendritic cells (DCs) [22]. In RA, TNF-α secreted from Th1 cells and macrophages activates synovial fibroblasts, promotes epidermal hyperplasia, and recruits inflammatory cells [34]. After activation by various cytokines, including IL-1, IL-6, and TNF-α [35], synovial fibroblasts overexpress cathepsins and matrix metalloproteinases (MMPs), followed by collagen and proteoglycan breakdown. As a result, cartilage and bone are destroyed, and finally, joint erosion occurs. Osteoclasts are also important for the progression of RA pathology throughout TNF-α activation, and activated osteoclasts in RA induce synovial hyperplasia and angiogenesis (Figure 2) [36].

### 2.2. Psoriatic Arthritis

Psoriatic arthritis (PsA) is a long-term autoimmune arthropathy distinct from RA [37]. It typically occurs in people affected by cutaneous psoriasis. Similar to RA, PsA is relatively common, affecting approximately 1% of the population [38]. The representative feature of PsA is the swelling of fingers and toes, leading to a sausage-like appearance [39]. The major cells involved in the PsA pathogenesis are DCs, macrophages, type 17 helper T cells (Th17), and keratinocytes [39,40]. DCs and macrophages activated by TNF-α produce TNF-α and IL-23 excessively. IL-23 causes differentiation of naïve T cells into Th17 cells, which overproduce IL-17. Then, IL-17 and TNF-α activate keratinocytes, which promote epidermal hyperplasia and recruit inflammatory cells such as DCs. TNF-α induces the proliferation and anti-apoptosis of keratinocytes via the NF-κB signal pathway, eventually leading to the formation of microabscess by enhancing the recruitment of inflammatory cells in psoriasis (Figure 2) [41].

### 2.3. Inflammatory Bowel Disease

Inflammatory bowel disease (IBD) is a group of autoimmune diseases localized in the gastrointestinal tract. CD and ulcerative colitis (UC) are the primary types of IBD, which are distinguished by the location at which the disease occurs. CD mostly occurs in the small and large intestine, but it can also affect the mouth, esophagus, stomach, and anus, while CD commonly affects the colon and the rectum [42,43]. In both CD and CD, TNF-α is secreted from Th1 cells along with other cytokines, including IL-1, IL-6, and IL-17 [44,45]. These cytokines are capable of accumulating intestinal fibroblasts, neutrophils, and macrophages in the gut. Accumulated intestinal fibroblasts cause intestinal fibrosis, which leads to the formation of stricture in the intestine. Accumulated neutrophils in the intestine secrete elastase to induce matrix degradation. Finally, accumulated macrophages in the intestine produce TNF-α, IL-1, and IL-6, which eventually induce intestinal matrix degradation, epithelial damage, endothelial activation, and vascular disruption (Figure 3) [23,24,25].

### 2.4. Psoriasis

Psoriasis (PS) is a chronic inflammatory disease with a strong genetic diathesis and autoimmunity characterized by a specific region of abnormal skin. There are five types of PS-laque, guttate, inverse, pustular, and erythrodermic, and about 90% of PS is plaque-type PS. Plaque-type PS classically presents pruritic plaques with white scales on the top [46,47]. The lesions of PS reflect the epidermal hyperplasia, angiogenesis, and inflammatory infiltration of leukocytes into the dermis as a result of the dysregulation of skin immune responses [48]. The inflammatory pathways involved in the pathogenesis of plaque psoriasis overlap the rest of the clinical variants [46]. Stressed keratinocytes excessively secreted TNF-α along with IL-1 and IL-6. Those cytokines activate DCs. IL-12 from activated DCs differentiates naïve T cells into Th1 cells, and IL-23 differentiates naïve T cells into Th17 cells [40]. These two kinds of cells overproduce specific cytokines, and Th1 cells secrete TNF-α and IFN-γ, whereas Th17 cells secrete IL-17 immoderately [46]. This abnormal immune reaction makes TNF-α activate DCs continuously. TNF-α, IFN-γ, and IL-17 ultimately lead to keratinocyte hyperproliferation and epidermal change, such as acanthosis, parakeratosis, and hypogranulosis (Figure 4) [40].

### 2.5. Noninfectious Uveitis

Noninfectious uveitis (NIU), also known as autoimmune uveitis, is an autoimmune disease inside the eye that aims at the neuroretina [49]. NIU can be associated with other autoimmune diseases, including RA, IBD, and PS. Patients with NIU present an expanded risk of retinal detachment, cataracts, developing glaucoma, and visual distribution, along with blindness or low vision [50]. The start of NIU is an overproduced TNF-α from macrophages, as well as other various cytokines such as IL-6, IL-10, IL-12, and IL-23. DCs are activated TNF-α and other cytokines [51,52]. Th1 cells are differentiated from naïve T cells by immoderate IL-12, which activated DCs produce. Th17 cells are differentiated by excessively secreted IL-6 and TGF-β from DCs. Activated Th1 cells and Th17 cells migrate and infiltrate inside the uvea, which is the middle layer of the eye and provides blood to the retina [53]. Migrated Th1 cells and Th17 cells generate damage to the blood–retinal barrier, trigger the retinal vasculature, and recruit nonspecific blood-circulating leukocytes such as monocytes, lymphocytes, and polymorphonuclear leukocytes (PMNs). The inflammation that makes swelling and destroys uvea eventually occurs in NIU (Figure 5) [54].

## 3. TNF-α Inhibitors as Therapeutic Drugs

### 3.1. Infliximab (Remicade^®^)

Infliximab is a recombinant chimeric monoclonal antibody containing a murine variable region and a human IgG1 constant region. It is specific for all forms of TNF-α in humans and effectively blocks the binding of TNF-α to its soluble and transmembrane receptors [55]. After infliximab treatment for IBD patients, the lysis of cell lines expressing TNF-α by the complement-dependent and antibody-dependent cytotoxicity was promoted and reduced inflamed tissue [56]. In addition to generating apoptosis, infliximab blocks IFN-γ production in colonic and stimulated T cells, and anti-inflammation occurs [57]. Furthermore, infliximab downregulates intracellular adhesion molecule 1 (ICAM-1) and vascular cell adhesion molecule 1 (VACM-1) and modulates the MMPs/tissue inhibitor of metalloproteinases (TIMPs) balance [58,59]. The half-life of infliximab is approximately 8–10 days and can be maintained by administering doses every eight weeks [60]. Infliximab was initially FDA approved for the therapy of CD in 1998 and later also for RA. It was further approved for the treatment of ankylosing spondylitis (AS), PsA, CD, and PS (Figure 6) [61].

### 3.2. Etanercept (Enbrel^®^)

Etanercept is a fusion protein that includes two identical TNFR2 extracellular regions connected to the Fc fragment of human IgG1 [62]. There are diverse variants of each etanercept group with very few differences [63]. Etanercept binds to sTNF-α or tmTNF-α and inactivates them by blocking the interaction with receptors [56]. It only enables it to bind to the active trimeric TNF-α since its binding site is placed in the cleft between subunits [64]. Etanercept has a half-life of 3–3.5 days after subcutaneous injection [60]. It relieves symptoms of arthritis and prevents the progression of RA in patients [65]. Etanercept also appears to modulate the proinflammatory genes such as NF-κB in plaque PS, resulting in a significant decrease in the production of TNF-α. Furthermore, it causes the apoptosis of large amounts of DCs in plaque, which stops positive feedback related to TNF-α by early apoptotic cell death before the activation and maturation of DCs [66]. It was permitted by FDA in 1998 for the treatment of juvenile idiopathic arthritis (JIA) and PsA, RA, PS, and AS (Figure 6) [67]. For articular involvement, favorable results have been reported, but there is a limited outcome in ocular inflammation such as uveitis [68].

### 3.3. Adalimumab (Humira^®^)

Adalimumab is a fully human IgG1 monoclonal antibody capable of specifically blocking the binding of human TNF-α to the receptors because its function and structure are identical to that of natural human IgG1 [69]. Adalimumab needs a less frequent subcutaneous administration because of its comparatively longer half-life, which is about 10 to 13 days [70]. Furthermore, adalimumab shows lower immunogenicity than infliximab [71]. Because adalimumab has better tolerance and lower immunogenicity, it is also effectively used for patients with CD and can be administered to patients who have had an allergic reaction to infliximab. In PA patients treated with adalimumab, declines in the levels of TNF-α and IL-6, as well as acute-phase reactants of inflammation, are observed [72]. Additionally, it was found that the treatment of RA with adalimumab can inhibit IL-17, which is oversecreted by Th17 cells by increasing the number of regulatory T cells (Treg) compared to untreated patients [73]. Adalimumab was approved by the FDA in 2002 for the treatment of RA, followed by approval for the treatment of PsA, PS, AS, JIA, and CD, as well as uveitis (Figure 6) [71,74].

### 3.4. Certolizumab Pegol (Cimzia^®^)

Certolizumab pegol (CDP87) is a humanized monoclonal antibody that has a polyethylene glycosylated (PEG) Fab fragment and lacks the Fc region [75]. Due to the lack of the Fc region, it does not include complement or antibody-dependent cytotoxicity. It is a novel TNF-α inhibitor with a remarkable mechanism of action in comparison to other TNF-α inhibitors. Because of the PEGylation, certolizumab pegol can be more significantly distributed into inflamed tissues compared to infliximab and adalimumab [76]. The distinct structure of the inhibitor might be the reason behind the higher efficacy of certolizumab pegol in comparison with other TNF-α inhibitors [77]. The PEGylation of certolizumab pegol enhances its half-life to two weeks, which may be attributed to the high concentration of the inhibitor examined in inflamed tissues [78]. Certolizumab pegol was approved for the treatment of CD, RA, PsA, As, and plaque PS by the FDA by 2018 (Figure 6) [79].

### 3.5. Golimumab (Simponi^®^we)

Golimumab is a completely humanized IgG1 monoclonal antibody that has functional specificity for human TNF-α [80]. Compared to infliximab and adalimumab, golimumab has a higher affinity and is more potent in the neutralization of sTNF-α and tmTNF-α, thus it can effectively inhibit the biological activity of TNF-α [81]. It also blocks the leukocyte infiltration by prevention of cell adhesion proteins such as E-selectin, ICAM-1, and VCAM-1, as well as proinflammatory cytokine secretion [82]. Golimumab’s half-life is about 7–20 days [83]. Golimumab was approved by the FDA in 2009 for the treatment of RA when injected in a mixture with methotrexate. It was also approved for UC, AS, and PsA treatment (Figure 6) [84].

### 3.6. TNF-α Inhibitor Biosimilars

Since the initial approval of each TNF-α inhibitor, several kinds of biosimilars that are not generic products but are highly similar to the originators have received FDA approval. These have only minor differences in clinically inactive components in the molecular structure, which are about the same in terms of purity, safety, and efficacy [85]. While biosimilars for infliximab, etanercept, and adalimumab have received FDA approval within the last five years, no biosimilars approved for certolizumab pegol and golimumab to date [86]. However, out of 12 approved biosimilars, only infliximab-axxq (Avsola^®^), infliximab-abda (Renflexis^®^), and infliximab-dyyb (Inflectra^®^) with infliximab as a reference product are launched and available in the US market (Table 1) [87].

### 3.7. New Anti-TNF-α Agents 

New anti-TNF-α agents are currently being developed to overcome shortcomings such as the nonresponse of existing TNF-α inhibitors. Ozoralizumab is a humanized monoclonal antibody designed for the treatment of inflammatory diseases. In a Phase 2 study, clinical activity and safety were evaluated in patients worldwide and in Japan. When 80 mg of ozoralizumab was injected, effective and well-tolerated treatment was possible as a novel TNF-α inhibitor (NCT01007175). Due to the specific molecular characteristics of the nanobody (small size, low immunogenicity potential, manufacturability), most patients showed remarkable improvement in disease activity, and once induced, remission can be maintained at a dose of less than 80 mg per month, which shows the desired treatment result. In addition, since Phase 3 is currently in progress, ozoralizumab is highly likely to be used as a new TNF-α inhibitor. 

Saddala and Huang established a new drug model, ZINC09609430, that could specifically inhibit TNF-α. It was screened based on the Zinc library using structure-based pharmacophore modeling, virtual screening, and molecular docking with in silico absorption, distribution, metabolism, excretion, and toxicity (ADMET) analysis. As a result, the identified new small compound could serve as structurally various novel inhibitors for TNF-α and could potentially be used as anti-inflammatory agents to treat related diseases. Although ZINC09609430 is safe and active in silico ADMET results, it is worthy of further evaluation for its safety and efficacy in vitro and in vivo [88].

TNF blocking treatment for CD is safe and effective, but due to the high nonresponse rate in patients, Kwak et al. conducted an in silico study to suggest a novel drug treatment for anti-TNF refractory CD. Among them, CP-69033401 showed high drug response in the cluster of patients with anti-TNF refractory CD. These findings could provide new options for anti-TNF treatment (Table 2) [89].

## 4. Conclusions and Discussion

Here, we introduced the association of TNF-α signaling in some autoimmune diseases and the current TNF-α inhibitors used as therapeutic drugs for these diseases. TNF-α, which is commonly known as a proinflammatory cytokine, has also been shown to apply pleiotropic effects on various cell models and has been identified as a significant factor in the pathogenesis of autoimmune diseases (Figure 7). The role of TNF-α in these diseases has not been entirely understood; however, it is generally known to contribute to the progression of disease when excessively produced by activating and accumulating fibroblasts, causing joint erosion, fibrosis, and stricture formation. Presently, five TNF-α inhibitors are clinically utilized to treat several inflammatory diseases. These include infliximab, adalimumab, etanercept, certolizumab pegol, and golimumab, which are monoclonal antibodies, and, lastly, etanercept, which is a fusion protein. These inhibitors are specific for all forms of TNF-α in humans, and effectively block the binding of TNF-α to its receptors. Compared with the earlier TNF-α inhibitors, golimumab is a monoclonal antibody that has a higher affinity to TNF-α and can neutralize the biological activity of TNF-α effectively.

Regarding safety issues, there are several side effects of TNF-α inhibitors. General side effects of all TNF-α inhibitors involve headaches, rashes, anemia, transaminitis, infectious, and, most commonly, an injection site reaction through a subcutaneous route [90,91]. The worsening of heart failure and increased risk of developing tuberculosis as well as lymphoma are significantly observed in patients who have autoimmune diseases compared to healthy people [92,93,94]. Thus, it is recommended that all individuals be screened for these before beginning the treatment, and preventive therapy must be performed by priority if they are detected.

Currently, more TNF-α signaling inhibitors for autoimmune diseases are being clinically assessed, and even biosimilars of approved therapeutic drugs are being developed by many research institutes and companies, as the market for TNF-α inhibitors shows great promise. Furthermore, TNF-α inhibitors are used in off-label indications as well as the approved therapies for autoimmune diseases. Case reports have demonstrated that infliximab, etanercept, and adalimumab are effective against diseases such as Pyoderma gangrenous, Granuloma annulare, and Pityriasis rubra pilaris [95]. Although studies on these are still lacking, it suggests that the applicability of the TNF-α inhibitors could be expanded.

Therefore, it can be predicted that the understanding of TNF-α signaling will be much more emphasized in the near future to develop effective tools for the treatment of other autoimmune diseases as well as a wide range of diseases involving TNF.

## Figures and Tables

**Figure 1 ijms-22-02719-f001:**
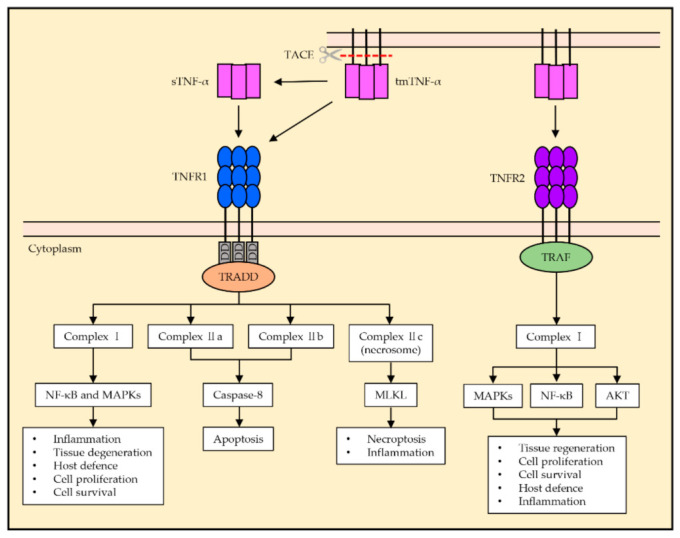
General tumor necrosis factor alpha (TNF-α) signaling pathway of TNFR1 and TNFR2. The TNFR1 signaling pathway is activated by the ligation of sTNF-α and tmTNF-α, and the death domain of TNFR1 recruits TRADD. Complex I activates NF-κB and MAPKs, which results in inflammation, tissue degeneration, host defense, cell proliferation, and cell survival. Complexes IIa and IIb activate caspase-8 and induce apoptosis. Complex IIc is known to induce necroptosis and inflammation via the activation of MLKL. The TNFR2 signaling pathway is mainly activated by tmTNF-α. TNFR2 does not possess a death domain and recruits TRAF via its TRAF domain, which activates the formation of complex I, resulting in NF-κB and MAPKs and AKT activation. TNFR2 activation is associated with homeostatic bioactivities such as tissue regeneration, cell proliferation, and cell survival, as well as host defense and inflammation.

**Figure 2 ijms-22-02719-f002:**
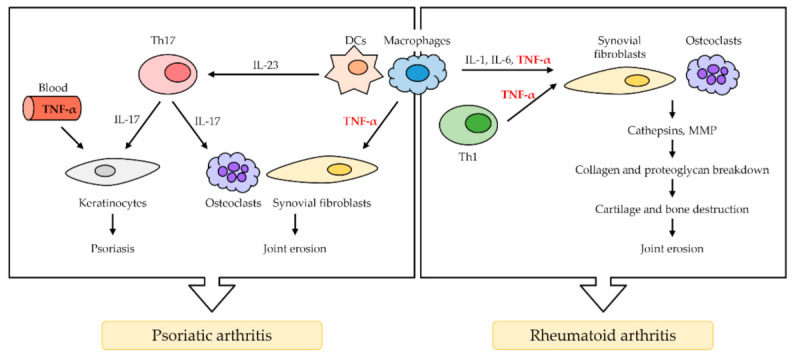
The role of TNF-α in rheumatoid arthritis (RA) and psoriatic arthritis (PsA). TNF-α is mainly secreted by macrophage and Th1 cells. In RA, TNF-α activates synovial fibroblasts, which causes the overproduction of cathepsins and MMP. The breakdown of collagen and proteoglycan follows, resulting in cartilage and bone destruction, as well as joint erosion. Osteoclasts in RA induce synovial hyperplasia and angiogenesis. In PsA, activated dendritic cells (DCs) and macrophages secrete TNF-α and IL-23 excessively. IL-23 induces T cells to differentiate in Th17 cells, which secretes IL-17. IL-17 and TNF-α in the blood activate keratinocytes, resulting in psoriasis (PS). TNF-α also activates osteoclasts, leading to synovial fibroblast, which results in joint erosion.

**Figure 3 ijms-22-02719-f003:**
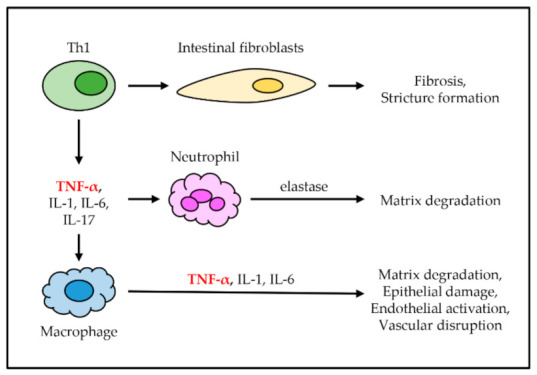
The role of TNF-α in inflammatory bowel disease (IBD). TNF-α is secreted from Th1 cells along with other cytokines. These cytokines cause the accumulation of immune cells, including intestinal fibroblasts, neutrophils, and macrophages in the gut. Intestinal fibroblasts cause fibrosis and stricture formation. Neutrophils secrete elastase, which causes intestinal matrix degradation. Macrophages produce more inflammatory cytokines, which causes intestinal matrix degradation, epithelial damage, endothelial activation, and disruption.

**Figure 4 ijms-22-02719-f004:**
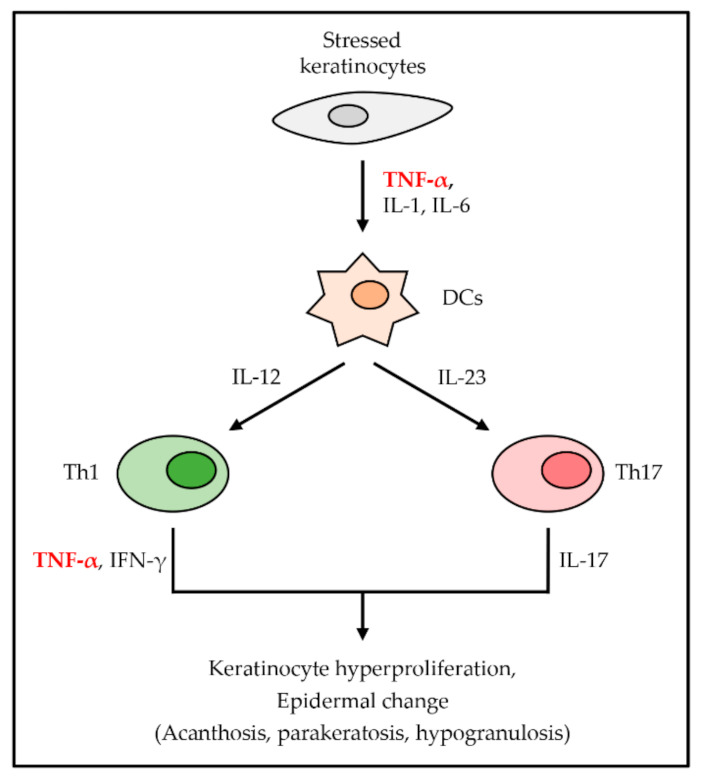
The role of TNF-α in PS. TNF-α is first secreted from stressed keratinocytes with other proinflammatory cytokines. Secreted cytokines activate DCs, which induce the differentiation of T cells into Th1 cells and Th17 cells. They secrete TNF-α, IFN-γ, and IL-17, respectively, which cause keratinocyte hyperproliferation and several epidermal changes.

**Figure 5 ijms-22-02719-f005:**
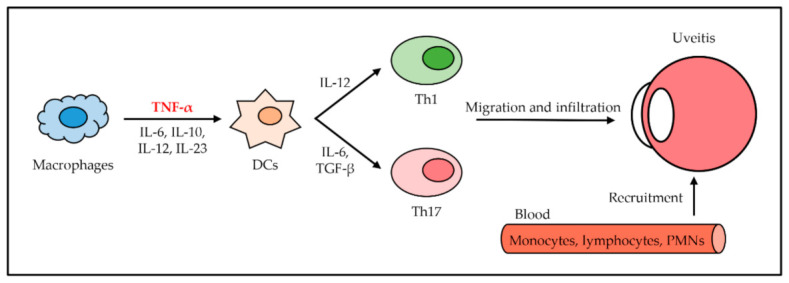
The role of TNF-α in uveitis. TNF-α is secreted from macrophages with other cytokines and activates DCs. Activated DCs secrete IL-12, IL-6, and TGF-β to induce differentiation of Th1 cells and Th17 cells from naïve T cells, respectively. Th1 cells and Th17 cells arrive at the uvea through migration and infiltration. Several leukocytes from blood are recruited by Th1 cells and Th17 cells. NIU occurs due to inflammation.

**Figure 6 ijms-22-02719-f006:**
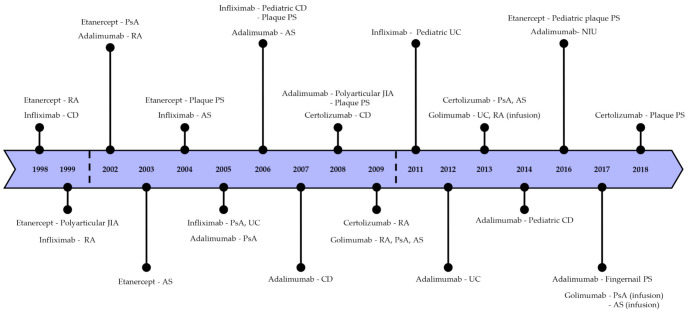
Timeline of the approval dates of TNF-α inhibitors in autoimmune diseases.

**Figure 7 ijms-22-02719-f007:**
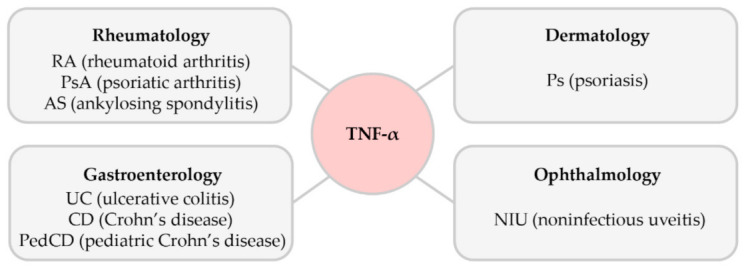
The association of TNF-α in various autoimmune diseases.

**Table 1 ijms-22-02719-t001:** TNF-α inhibitor originators and FDA-approved biosimilars by the end of 2020.

Originators (Reference Products)	Biosimilars (FDA-Approved Products)	Approval Date	Launch Date (US)
Infliximab (Remicade^®^)	Infliximab-axxq (Avsola^®^)	December 2019	July 2020
	Infliximab-qbtx (Ixifi^TM^)	December 2017	Not launching in US
	Infliximab-abda (Renflexis^®^)	May 2017	July 2018
	Infliximab-dyyb (Inflectra^®^)	April 2016	November 2016
Etanercept (Enbrel^®^)	Etanercept-ykro (Eticovo^TM^)	April 2019	-
	Etanercept-szzs (Erelzi^®^)	August 2016	-
Adalimumab (Humira^®^)	Adalimumab-fkjp (Hulio^TM^)	July 2020	-
	Adalimumab-afzb (Abrilada^TM^)	November 2019	-
	Adalimumab-bwwd (Hadlima^TM^)	July 2019	-
	Adalimumab-adaz (Hyrimoz^®^)	October 2018	-
	Adalimumab-adbm (Cyltezo^®^)	August 2017	-
	Adalimumab-atto (Amjevita^TM^)	September 2016	-
Certolizumab pegol (Cimzia^®^)	No biosimilars approved		
Golimumab (Simponi^®^)	No biosimilars approved		

**Table 2 ijms-22-02719-t002:** New anti-TNF-α agents currently under development.

Drugs	Research institute	Form	Indications	Progress	Ref.
Ozoralizumab (TS-152)	Taisho	Monoclonal antibody	RA	Phase 3 (NCT04077567) Est. End Date: Dec. 2022	-
ZINC09609430	University of Missouri	Small-molecule drug compound	Unknown	In silico study	[88]
CP-690334-01	Kyung Hee University College of Medicine	Small-molecule drug compound	CD	In silico study	[89]

## Data Availability

Not applicable.

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
