# Peer review of "The Role of Tumor Necrosis Factor Alpha (TNF-α) in Autoimmune Disease and Current TNF-α Inhibitors in Therapeutics"

_ijms, 2021, doi:10.3390/ijms22052719_

Round 1

Reviewer 1 Report

The authors provide a well structured overview about the role of TNFalpha in autoimmune diseases and its therapeutic agents.

Major:

In the discussion it is mentioned that there are new anti TNFalpha agents under development. I would recommend to include an overview (eg in a table) directly as a subchapter after the approved five biologics.

In addition I am wondering about the fact that the names of the original drug are given, but I guess that there are also Biosimilars available on the market? Maybe this should be mentioned in the text.

Furthermore I am missing a short paragraph on other indications where TNFalpha blockers are or can be used.

Minor:

Line 50: moved in between figure 1 and its caption

Figure 1: please correct typo in “CAPASE” = “CASPASE”

Line 109: I guess what is meant is “blocking … interaction”

Author Response

Point-by-Point Responses

Reviewer‘ comments:

Reviewer #1:

Above all, we would like to express our genuine gratitude for the time and effort which had put into reviewing our manuscript and providing insightful comments. All authors have cautiously reviewed the comments in detail and revised the manuscript with care.

Major:

In the discussion it is mentioned that there are new anti TNFalpha agents under development. I would recommend to include an overview (eg in a table) directly as a subchapter after the approved five biologics.

We really appreciate your detailed comments. Reviewer #2 also pointed out this issue. We thought that additional explanations are needed for the new anti-TNF-α agents under development, so after explaining 5 biologics, we added another section, ‘3.7. Novel anti-TNF-α agents’ as follows:

3.7. New anti-TNF-α agents

New anti-TNF-α agents to overcome the shortcomings such as non-response of existing TNF-α inhibitors is currently being developed. Ozoralizumab is a humanized monoclonal antibody designed for the treatment of inflammatory diseases. In a Phase 2 study, clinical activity and safety were evaluated in patients worldwide and in Japan. When 80mg of ozoralizumab was injected, effective and well-tolerated treatment was possible as a novel TNF-α inhibitor (NCT01007175). Due to the specific molecular characteristics of the nanobody (small size, low immunogenicity potential, manufacturability), most patients showed remarkable improvement in disease activity, and once induced re-mission can be maintained at a dose of less than 80mg per month, it shows the desired treatment result. In addition, since Phase 3 is currently in progress, ozoralizumab is highly likely to be used as a new TNF-α inhibitor.

Saddala MS & Huang H. established a new drug model, ZINC09609430, that could specifically inhibit TNF-α. It was screened based on the Zinc library using structure-based pharmacophore modeling, virtual screening, and molecular docking with in silico AD-MET (absorption, distribution, metabolism, excretion, and toxicity) analysis. As a result, the identified new small compound could serve as structurally various novel inhibitors for TNF-α, and could potentially be used as anti-inflammatory agents to treat related dis-eases. Although ZINC09609430 is safe and active in silico ADMET results, it is worth further evaluation for its safety and efficacy in vitro and in vivo [88].

TNF blocking treatment for CD is safe and effective, but due to the high non-response rate in patients, Kwak MS et al. conducted an in silico study to suggest a novel drug treatment for anti-TNF refractory CD. Among them, CP-69033401 showed high drug response in the cluster of patients with anti-TNF refractory CD. These findings could provide new options for anti-TNF treatment [89].

Table 2. New anti-TNF-α agents currently under development.

Drugs

Research institute

Form

Indications

Progress

Ref.

Ozoralizumab (TS-152)

Taisho

Monoclonal antibody

RA

Phase 3 (NCT04077567) Est. End Date: Dec. 2022

-

ZINC09609430

University of Missouri

Small-molecule

drug compound

Unknown

In silico study

[88]

CP-690334-01

Kyung Hee University College of Medicine

Small-molecule

drug compound

CD

In silico study

[89]

In addition I am wondering about the fact that the names of the original drug are given, but I guess that there are also Biosimilars available on the market? Maybe this should be mentioned in the text.

We sincerely appreciate your in-depth comment to make our manuscript better. Following your advice, we added a section on the ‘3.6. TNF-α inhibitor biosimilars’ to mention the latest trend in biosimilars and summarized in Table 1. Its contents were added to line 279-289 of the manuscript as follows:

3.6. TNF-α inhibitor Biosimilars

Since the initial approval of each TNF-α inhibitor, several kinds of biosimilars that are not generic products but are highly similar to the originators have received FDA approvals. These have only minor differences in clinically inactive components in the molecular structure and are about the same in terms of purity, safety, and efficacy [84]. While biosimilars for infliximab, etanercept, adalimumab has received FDA approval within the last 5 years, no biosimilars approved for certolizumab pegol and golimumab to date [85]. However, out of 12 approved biosimilars, only infliximab-axxq (Avsola®), infliximab-abda (Renflexis®) and infliximab-dyyb (Inflectra®) with infliximab as a reference product are launched and available in the US market (Table 1) [86].

Table 1. TNF-α inhibitor originators and FDA-approved biosimilars by the end of 2020.

Originators

(Reference products)

Biosimilars

(FDA- approved products)

Approval date

Launch date

(US)

Infliximab (Remicade®)

Infliximab-axxq (Avsola®)

Dec 2019

Jul 2020

Infliximab-qbtx (IxifiTM)

Dec 2017

Not launching in US

Infliximab-abda (Renflexis®)

May 2017

Jul 2018

Infliximab-dyyb (Inflectra®)

Apr 2016

Nov 2016

Etanercept (Enbrel®)

Etanercept-ykro (EticovoTM)

Apr 2019

-

Etanercept-szzs (Erelzi®)

Aug 2016

-

Adalimumab (Humira®)

Adalimumab-fkjp (HulioTM)

Jul 2020

-

Adalimumab-afzb (AbriladaTM)

Nov 2019

-

Adalimumab-bwwd (HadlimaTM)

Jul 2019

-

Adalimumab-adaz (Hyrimoz®)

Oct 2018

-

Adalimumab-adbm (Cyltezo®)

Aug 2017

-

Adalimumab -atto (AmjevitaTM)

Sep 2016

-

Certolizumab pegol (Cimzia®)

No biosimilars approved

Golimumab (Simponi®)

No biosimilars approved

Furthermore I am missing a short paragraph on other indications where TNFalpha blockers are or can be used.

We are grateful for your valuable comments. Based on your comments, we further added short paragraph on other indications for TNF-α inhibitors to the Conclusion and Discussion section as follows:

Example:

  1. “Furthermore, TNF-α inhibitors are also used in off-label indications as well as the approved therapies for autoimmune diseases. Case reports have demonstrated that infliximab, etanercept, and adalimumab are effective against diseases such as Pyoderma gangrenous, Granuloma annulare, and Pityriasis rubra pilaris [68]. Although studies on these are still lacking, it suggests that the applicability of the TNF-α inhibitors could be expended.” (line 344-349 of the manuscript)

Minor:

Line 50: moved in between figure 1 and its caption

Figure 1: please correct typo in “CAPASE” = “CASPASE”

Line 109: I guess what is meant is “blocking … interaction”

Thank you for pointing out our mistakes. We corrected that you mentioned in minor points and also carefully overviewed the whole paper to make sure there were no other errors.

Reviewer 2 Report

In this review, the authors summarize the association of TNF-α signaling in some autoimmune diseases, and the current TNF-α inhibitors used as therapeutic drugs for these diseases. Considering the quality of this manuscript, I would like to recommend it for publication after some minor revisions. The detailed comments are as follows:

  1. This review cites few examples for the pathology of TNF-α signaling in autoimmune diseases, I think the author should consider adding more examples.
  2. This review lacks a specific description of the current outstanding works in TNF-α inhibitors as therapeutic drugs section. I think the author should select some therapeutic drugs for careful analysis.
  3. This review lacks the evaluation of its effects in the TNF-α inhibitors as therapeutic drugs section. I think the author should conduct a more comprehensive and detailed evaluation of its effects in the TNF-α inhibitors as therapeutic drugs section.
  4. I think this review should give a more detailed description of TNF-α limitations and prospects in the conclusion section.
  5. I recommend that the author check whether the abbreviations in the text have been given their full names as the first time they appear, or consider making a list of abbreviations.
  6. The English can be improved by a native speaker.

Author Response

Point-by-Point Responses

Reviewer‘ comments:

Reviewer #2:

Above all, we would like to express our genuine gratitude for the time and effort which had put into reviewing our manuscript and providing insightful comments. All authors have cautiously reviewed the comments in detail and revised the manuscript with care.

This review cites few examples for the pathology of TNF-α signaling in autoimmune diseases, I think the author should consider adding more examples.

Thank you very much for your comments. We should have added more examples on the pathology of TNF-α signaling in autoimmune diseases. Thus, we added examples of other autoimmune disease related to TNF-α signaling in the ‘2.4 Psoriasis’ and ‘2.5 Non-infectious uveitis’ sections. The addition is as follows:

  1. 4. Psoriasis

Figure 4. The role of TNF-α in PS. TNF-α is first secreted from stressed keratinocytes with other pro-inflammatory cytokines. secreted cytokines activate DCs, and DCs induce differentiation of T cells into Th1 cells and Th17 cells. They secrete TNF-α, IFN-γ, and IL-17 respectively, which cause keratinocyte hyperproliferation and several epidermal changes.

Psoriasis (PS) is a chronic inflammatory disease with a strong genetic diathesis and autoimmunity characterized by a specific region of abnormal skin. There are five types of PS - plaque, guttate, inverse, pustular, and erythrodermic and about 90% of PS is plaque-type PS. The plaque type PS classically presents pruritic plaques with white scales on top [46, 47]. The lesions of PS reflect epidermal hyperplasia, angiogenesis, and inflammatory infiltration of leukocytes into the dermis as a result of a dysregulation of skin immune responses [48]. The inflammatory pathways involved in the pathogenesis of plaque psoriasis overlap the rest of clinical variants [46]. Stressed keratinocytes excessive-ly secreted TNF-α along with IL-1 and IL-6. Those cytokines make DCs activated. IL-12 from activated DCs differentiates naïve T cells into Th1 cells, and IL-23 differentiates naïve T cells into Th17 cells [40]. These two kinds of cells overproduce specific cytokines, Th1 cells secrete TNF-α and IFN-γ, whereas Th17 cells secrete IL-17 immoderately [46]. This abnormal immune reaction makes TNF-α activates DCs continuously. TNF-α, IFN-γ, and IL-17 ultimately lead to keratinocyte hyperproliferation and epidermal change, such as acanthosis, parakeratosis, and hypogranulosis (Figure 4) [40].

  1. 5. Non-infectious uveitis

Figure 5. The role of TNF-α in uveitis. TNF-α is secreted from macrophages with other cytokines and activates DCs. Activated DCs secrete IL-12, IL-6, and TGF-β to induce differentiation of Th1 cells and Th17 cells from naïve T cells respectively. Th1 cells and Th17 cells arrive at the uvea through migration and infiltration. Several leukocytes from blood are recruited by Th1 cells and Th17 cells. NIU occurs due to inflammation.

Non-infectious uveitis (NIU), also known as autoimmune uveitis, is an autoimmune disease inside the eye which aims at the neuroretina [49]. NIU can be associated with other autoimmune diseases including RA, IBD, and PS. Patients with NIU appear an expanded risk of retinal detachment, cataracts, developing glaucoma, and visual distribution along with blindness or low vision [50]. The start of NIU is an overproduced TNF-α from macrophages as well as other various cytokines like IL-6, IL-10, IL-12, and IL-23. DCs are activated TNF-α and other cytokines [51, 52]. Th1 cells are differentiated from naïve T cells by immoderate IL-12 which activated DCs produce. Th17 cells are differentiated by excessively secreted IL-6 and TGF-β from DCs. Activated Th1 cells and Th17 cells migrate and infiltrate inside the uvea, which is the middle layer of the eye and provides blood to the retina [53]. Migrated Th1 cells and Th17 cells generate damage to the blood-retinal barrier, trigger the retinal vasculature, and recruit nonspecific blood-circulating leukocytes such as monocytes, lymphocytes, and polymorphonuclear leukocytes (PMNs). The inflammation that makes swelling and destroys uvea occurs NIU eventually (Figure 5) [54]

This review lacks a specific description of the current outstanding works in TNF-α inhibitors as therapeutic drugs section. I think the author should select some therapeutic drugs for careful analysis.

We appreciate your in-deep comments. Reviewer #1 also pointed out this issue. We thought that additional explanations are needed for the new anti-TNF-α agents under development, so after explaining 5 biologics, we added another section, ‘3.7. Novel anti-TNF-α agents’.

This review lacks the evaluation of its effects in the TNF-α inhibitors as therapeutic drugs section. I think the author should conduct a more comprehensive and detailed evaluation of its effects in the TNF-α inhibitors as therapeutic drugs section.

We sincerely apologize for our insufficient explanation regarding the effect of TNF-α inhibitors as therapeutic drugs in autoimmune diseases. So, we further expanded the paragraphs and rephrased them to make sure they deliver more reliable description for the effect of TNF-α inhibitors as therapeutic drugs in autoimmune diseases.

Example:

  1. “After infliximab treatment for IBD patients, the lysis of cell lines expressing TNF-α by the complement-dependent and antibody-dependent cytotoxicity was promoted and reduced inflamed tissue [56]. In addition to generating apoptosis, infliximab blocks IFN-γ production in colonic and stimulated T cells to occur anti-inflammation [57]. Furthermore, infliximab downregulates intracellular adhesion molecule 1 (ICAM-1) and vascular cell adhesion molecule 1 (VACM-1) and modulates MMPs/tissue inhibitor of metalloproteinases (TIMPs) balance [58, 59]. (line 214-220 of the manuscript)
  2. “Etanercept also appears to modulate the pro-inflammatory genes such as NF-κB in plaque PS, resulting in a significant decrease in the production of TNF-α. Furthermore, it causes apoptosis of large amounts of DCs in plaque, which stops positive feedback related to TNF-α by early apoptotic cell death before activation and maturation of DCs [66]” (line 233-237 of the manuscript)
  3. “In PA patients treated with adalimumab, declines in the levels of TNF-α and IL-6, as well as acute phase reactants of inflammation, are observed [72]. Besides, it was found that the treatment of RA with adalimumab can inhibit IL-17, which is over-secreted by Th17 cells by increasing the number of regulatory T cells (Treg) compared to untreated patients [73].” (line 249-253 of the manuscript)

I think this review should give a more detailed description of TNF-α limitations and prospects in the conclusion section.

Thanks for the comments that allow our manuscript to be better. we gave a biased explanation on the effectiveness of treatment with TNF-α inhibitors, and there was a lack of perspective on its limitations. So, we revised and added the Conclusion and Discussion section to include limitations for the side effect of the TNF-α inhibitors to as follow:

Example:

  1. “Regarding to safety issues, there are several side effects of TNF-α inhibitors. General side effects of all TNF-α inhibitors involve headaches, rashes, anemia, transaminitis, infectious, and most commonly, injection site reaction through subcutaneous route [90, 91]. Worsening of heart failure and increased risk of developing tuberculosis as well as lymphoma are significantly observed in patients who have autoimmune disease compared to the healthy person [92-94]. Thus, it is recommended that all individuals be screened for these before beginning the treatment, and preventive therapy must be performed by priority if they are detected.” (line 333-340 of the manuscript)

I recommend that the author check whether the abbreviations in the text have been given their full names as the first time they appear, or consider making a list of abbreviations.

The English can be improved by a native speaker.

We really appreciate your meticulous advice. Based on your comments, we compiled a list of abbreviations in our manuscript and added them as follows:

AKT

Protein kinase B

AS

Ankylosing spondylitis

CD

Crohn’s disease

cIAP1/2

Cellular inhibitor of apoptosis protein 1 or 2

DCs

Dendritic cells

DD

Death domain

FADD

Fas-associated protein with death domain

IBD

Inflammatory bowel disease

ICAM-1

Intracellular adhesion molecule 1

JIA

Juvenile idiopathic arthritis

LUBAC

Linear ubiquitin chain assembly complex

MAPKs

Mitogen-activated protein kinases

MLKL

Mixed lineage kinase domain like protein

MMP

Matrix metalloproteinases

NF-κB

Nuclear factor κB

NIU

Non-infectious uveitis

PEG

Polyethylene glycosylated

PMNs

Polymorphonuclear leukocytes

PS

Psoriasis

PsA

Psoriatic arthritis

RA

Rheumatoid arthritis

RIPK1/3

Receptor-interacting serine/threonine-protein kinase 1 or 3

sTNF-α

Soluble TNF-α

TACE

TNF-α-converting enzyme

Th1

Type 1 helper T cells

Th17

Type 17 helper T cells

TIMPs

Tissue inhibitor of metalloproteinases

tmTNF-α

transmembrane TNF-α

TNF-α

Tumor necrosis factor alpha

TNFR1

TNF receptor type 1

TNFR2

TNF receptor type 2

TRADD

TNFR1-associated death domain

TRAF2/5

TNFR-associated factor 2 or 5

Treg

Regulatory T cells

UC

Ulcerative colitis

VACM-1

Vascular cell adhesion molecule 1

And, one of authors in this manuscript, Hye-Yoon Shin, is an English native speaker who was born in the U.S.A. She has carefully read and checked the manuscript and corrected some grammar and collocation mistakes.

Reviewer 3 Report

The reviewer would like to thnk the authors for their efforts in writing down this review. During the reviewing process, some points had been raised and require correction or clarification.

INTRODUCTION

Line 36, cytokines, chemotactic cytokines, and chemokines: chemotactic cytokines is the same as chemokines. Please remove one of the terms.

Line 41-42, various biological activities through Type 1 (TNFR1, also known as TNFRSF1A, CD120a, 42 p55) and Type 2 ( receptors: It's preferable if you could write it as Type 1 receptors..... and type 2 receptors to make the sentence more understandable.

Figure 1: Please include a sentence in the paragraph referring to  figure one. Also please redraw the pathway showing the correct pathway for TNFR1 and TNFR2 as TRADD as TNFR1 acts mainly on TRADD while TNFR2 acts on  TRAF. Please Check?! Also add detailed labels on the diagram. Please combine figure 1 and 2 in one figure showing the difference between the TNFR1 and TNFR2 pathways as it would be much more easier for comparing the two pathways when they are side by side.

It would be also nice if you add a small table of comparison between TNFR1 and TNFR2, so the readership can get the difference quickly than reading through all these paragraphs and captions.

Line 93-105: Please rewrite this paragraph as it's repeating itself. Also remove any details about each disease as it's written in details in the next paragraphs.

Line 129-130, y cells, predominantly type 1 helper T (Th1) cells: better write the whole term then abbreviate it....Type 1 helper T cells (Th1).

Again Figure 3 and 4 can be summarized in one figure as one side of figure 4 is the repetition of figure 3.

Figure 6: The authors had summarized the involvement of TNF-a in different disease. Here they had mentioned its role in opthalmology but never mentioned anything about it in the text or any paragraph. They should include a paragraph about the role of TNF-a in opthalmology.

It would be great also if the authors would add a timeline for the approval of each TNF-a inhibitors to be easily seen by the readership.

Author Response

Point-by-Point Responses

Reviewer‘ comments:

Reviewer #3:

Above all, we would like to express our genuine gratitude for the time and effort which had put into reviewing our manuscript and providing insightful comments. All authors have cautiously reviewed the comments in detail and revised the manuscript with care.

Line 36, cytokines, chemotactic cytokines, and chemokines: chemotactic cytokines is the same as chemokines. Please remove one of the terms.

We apologize for not being careful with the description. For the part you mentioned, the expression “cytokines, chemotactic cytokines, and chemokines” has been modified to “cytokines and chemokines”.

Line 41-42, various biological activities through Type 1 (TNFR1, also known as TNFRSF1A, CD120a, 42 p55) and Type 2 ( receptors: It's preferable if you could write it as Type 1 receptors..... and type 2 receptors to make the sentence more understandable.

Thank you for your comments. As per your recommend, we clarified the meaning through correction as follow:

Example:

  1. “Type 1 (TNFR1, also known as TNFRSF1A, CD120a, p55) and Type 2 (TNFR2, also known as TNFRSF1B, CD120b, p75) receptors”

→ rephrased. “Type 1 receptors (TNFR1, also known as TNFRSF1A, CD120a, p55) and Type 2 receptors (TNFR2, also known as TNFRSF1B, CD120b, p75)” (line 44-46 of the manuscript)

Figure 1: Please include a sentence in the paragraph referring to figure one. Also please redraw the pathway showing the correct pathway for TNFR1 and TNFR2 as TRADD as TNFR1 acts mainly on TRADD while TNFR2 acts on TRAF. Please Check?! Also add detailed labels on the diagram. Please combine figure 1 and 2 in one figure showing the difference between the TNFR1 and TNFR2 pathways as it would be much more easier for comparing the two pathways when they are side by side.

It would be also nice if you add a small table of comparison between TNFR1 and TNFR2, so the readership can get the difference quickly than reading through all these paragraphs and captions.

We greatly appreciate that you have pointed out what could help us make our manuscript more refined. As you have commented, splitting figure 1 and figure 2 may make the readership uncomfortable comparing the two signaling pathway and the labeling of the figures looks ambiguous. Therefore, we combined the figures so that they can be seen at a glance and redrawn them with a new figure 1, clearly labeling each molecule. We also clarified the sentence referring figure 1 in the paragraph and corrected the figure legend.

Line 93-105: Please rewrite this paragraph as it's repeating itself. Also remove any details about each disease as it's written in details in the next paragraphs.

We appreciate your point from our manuscript. As your comment about to modify part of introduction. Therefore line 94 to 107 was changed for following paragraph. Anything detail about the disease are removed and rewrite to make clear.

  1. “Physiologically, TNF-α is crucial for the normal immune response to infection as a regulator which largely activates the immune system, however inappropriate or excessive production of TNF-α can be harmful and may lead to disease. Abnormal secretion of TNF-α is often observed in the autoimmune disease and such as rheumatoid arthritis (RA), inflammatory bowel disease (IBD), and psoriatic arthritis (PsA), suggesting that TNF-α can be identified as a key factor in the pathological development. RA and PsA cause erosive inflammation and substantial damage of joints, which results in pain, stiffness and swelling. TNF-α has been also shown to have multiple, significant effects in these diseases by inducing inflammatory response and activating synovial fibroblasts in joints and bone which leads to joint erosion and bone destruction [22, 23]. IBD is a group of intestinal disorders that are characterized by chronic inflammation of the digestive tract. In the pathogenesis of IBD, TNF-α induces the accumulation of macrophages and intestinal fibroblasts, eventually leading to stricture formation as well as gut and intestinal matrix degradation [24-26].”

→ Rephrased. “Physiologically, TNF-α is crucial component for the normal immune response. TNF-α could activate the immune system to regulate, however, inappropriate or excessive production of TNF-α can be harmful and may lead to disease. The rheumatoid arthritis (RA) [22], inflammatory bowel disease (IBD) [23-25], psoriatic arthritis (PsA), psoriasis (PS) [26], and non-infectious uveitis (NIU) are induced by abnormal secretion of TNF-α, thus, TNF-α can be classified as a key factor in the pathological development.” (line 93-98 of the manuscript)

Line 129-130, y cells, predominantly type 1 helper T (Th1) cells: better write the whole term then abbreviate it....Type 1 helper T cells (Th1).

Regarding the abbreviation you mentioned, we corrected it to “type 1 helper T cells (Th1)”. You can see the modification in line 44-46 of the manuscript.

Again Figure 3 and 4 can be summarized in one figure as one side of figure 4 is the repetition of figure 3.

We are very pleased with your helpful comments. Rheumatoid arthritis and psoriatic arthritis have in common that it occurs at the joints, but differences in the pathogenesis are clear. Taking this into account, we represented figure 3 and 4 and revised the figure legends.

Figure 6: The authors had summarized the involvement of TNF-a in different disease. Here they had mentioned its role in ophthalmology but never mentioned anything about it in the text or any paragraph. They should include a paragraph about the role of TNF-a in ophthalmology.

We deeply apologize for the lack of description of ophthalmology in figure 6. In addition, the Reviewer #2 point out that we need to add more examples for the pathology of TNF-a signaling in autoimmune disease. So, we added a new section ‘2.5 Non-infectious uveitis’ on non-infectious uveitis, an ophthalmic autoimmune disease.

It would be great also if the authors would add a timeline for the approval of each TNF-a inhibitors to be easily seen by the readership.

We are grateful for your comments on improving the readability of our manuscript. Based on your comments, we drew a new figure of the timeline for the approval dates of TNF-a inhibitors in each autoimmune disease in the ‘3. TNF-a inhibitors as therapeutic drugs’ section and has been referred in the appropriate sentence.

Round 2

Reviewer 1 Report

No further comments

Reviewer 2 Report

The authors have addressed most of the concerns. I recommend it can be accepted in this form.

Reviewer 3 Report

The reviewer would like to thank the authors for their efforts in modifying the manuscript. It was well corrected. Thank you again for your efforts.